# Multi-Coil FD-EMI in Tidal Flat Areas: Prospection and Ground Truthing at a 17th Century Wooden Ship Wreckage

**Dennis Wilken [1,*], Daniel Zwick [2], Bente Sven Majchczack [1,3], Ruth Blankenfeldt [4], Ercan Erkul [1], Simon Fischer [1] and Dirk Bienen-Scholt [5]**

1   Institute of Geosciences, Kiel University, Otto-Hahn-Platz 1, 24118 Kiel, Germany;
    bmajchczack@roots.uni-kiel.de (B.S.M.); ercan.erkul@ifg.uni-kiel.de (E.E.); simon.fischer@ifg.uni-kiel.de (S.F.)
2   State Archaeology Department of Schleswig-Holstein (ALSH), Brockdorff-Rantzau-Straße 70,
    24837 Schleswig, Germany; Daniel.Zwick@alsh.landsh.de
3   Cluster of Excellence ROOTS, Kiel University, Leibnizstraße 3, 24118 Kiel, Germany
4   Centre for Baltic and Scandinavian Archaeology (ZBSA), Schloss Gottorf, 24837 Schleswig, Germany;
    ruth.blankenfeldt@zbsa.eu
5   Community Office Hallig Hooge, Hanswarft 1, 25859 Hallig Hooge, Germany; dbs@hooge.de
*   Correspondence: dennis.wilken@ifg.uni-kiel.de

**Abstract:** We present a case study of multi-coil frequency-domain electromagnetic (FD-EMI) prospection of a wooden ship wreckage from the 17th century. The wreckage is buried in a sandbar in the German part of the tidal flat area of the North Sea. Furthermore, the wreckage was excavated in advance and covered again after investigation. This ground truthing background and the position of the wreckage makes it a unique investigation object to test the feasibility of FD-EMI for prospecting wooden archaeological objects in the high conductive sediments of tidal flat areas. Our results reveal the shape and position of the wreckage in terms of conductivity maps. The resulting signal change caused by the wreckage in conductivity is only 10% of the value of the water-saturated sandy background, respectively, making a cautious process necessary, including a precise height correction. The data, furthermore, reveals a sensitivity to the vertical shape of the wreckage and thus sufficient depth sensitivity, but with reduced sensing depth. The study highlights the great potential of EMI for both in situ heritage management and archaeological research in the Wadden Sea.

**Keywords:** archaeological prospection; geophysics; EMI; tidal flats; ship wreck

## 1. Introduction

The Wadden region along the North Sea coast is an area of dyked or formerly dyked salt marshes and reclaimed coastal peat bogs [1]. This coastal wetland contains visible, past human adaptations to the environment in the form of embankments, dykes, canals, and polders, making it a cultural landscape of exceptional cultural, historical value ([1,2]). Natural and human-influenced dynamics have changed the marshes and tidal flats throughout time. These changes are especially visible by the numerous traces of medieval and post-medieval settlements and remains of their cultural landscapes that appear and disappear in the ever-changing environment of the Wadden Sea. Usually, archaeological site investigation in the tidal flat region of Germany has been performed using boreholes, trenches, surface findings, and (aerial) photographs. Visual surveying of archaeological traces and surface findings followed storm surges that removed sediments covering archaeological targets. However, sediment coverage of such remains is either restored by sedimentation/sediment transport from moving tidal creeks, and the tidal wave itself or the cultural remains are eroded by these [3]. Archaeological traces often remain or again become buried. Thus, feasible non-invasive subsoil/geophysical prospection methods are desirable to image and monitor archaeological sites. Feasible prospection means applying methods that fulfill the following demands:

1.  low weight to be able to carry devices in tidal flats over large distances to access remote areas of archaeological interest,
2.  fast and independent real-time surveying to cope with the short, low tide time window,
3.  sufficient contrast for wooden archaeological structures,
4.  and sufficient depth penetration.

Especially fast, area-wise operating (mapping) methods with the capability of providing depth information are needed, not only because of the short available time window during low tide, but also to develop monitoring concepts. As tidal creek erosion constantly moves sediments, endangering the cultural remains, a feasible monitoring method is even more important in the Wadden Sea region. Several geophysical techniques are not feasible, yet the saline character of the intertidal zone is again a restricting factor [4]. For example, ground-penetrating radar has a severely reduced depth penetration in saline/brackish wetland environments because of signal damping (e.g., [5]). Magnetic gradiometry allows fast measurement progress but does not provide any depth information if not combined with other methods. DC geoelectric methods, such as electrical resistivity tomography (ERT), are of course applicable to tidal flat and shallow water areas ([6,7]) but not feasible for area-wise prospection because of the laborious measurement setup. Furthermore, large changes in conductivity over the cycle of a day due to tidal water level change can be expected in the tidal flat region. This means that we can expect changes in conductivity during a tomographic measurement, resulting in time-dependent results. Usually, seismic/hydroacoustic methods provide a good combination of efficiency, resolution, contrast, and depth penetration (e.g., [8–11]) but most available (3D) systems have problems in very shallow waters in terms of water depth, water column multiples, weather conditions, and in areas that due to their topographic height only have a very short period of flooding during high tide. Nevertheless, seismic methods are a feasible expansion of the tidal flat prospection during high-tide periods.

The application of electromagnetic induction (EMI) methods promises to provide several benefits concerning the above-listed demands. EMI probably shows noteworthy contrast in terms of conductivity for archaeological wooden objects [12] and has the potential to even access their state of preservation, as wood physical properties significantly change with degradation and water saturation [8]. The method also provides suitable spatial resolution and depth penetration for imaging archaeological objects and has been successfully applied for archaeological prospection and palaeo-landscapes' reconstruction in several geological backgrounds (e.g., [13–16]). Nevertheless, following [4] the viability of EMI data in the intertidal zone is doubtful when using the low-induction-number (LIN) approximation [17] to calculate the apparent conductivities from the quadrature measurement values because of the non-linearity of the problem in high conductive areas [18] and a resulting limited dynamic range. The data might thus be biased (true conductivities are increasingly underestimated), and depth sensitivities change/are difficult to estimate without proper calibration. Ref. [4] performed measurements in the intertidal zone directly at the Belgium coastline, which allowed the application of a broad methodological setup, including calibration measurements and ground truthing based on corings and cone penetrations tests. They showed that the depth of exploration for EMI measurements is reduced by about 40%, a LIN-approximation bias, though a simple correction was applicable for their EMI device (correction coefficients were provided by [19], and that a change in salinity and groundwater level due to the tidal wave needs caution. The interpretation of the in-phase component in regions of high electrical conductivity is complex. Salty, clayey soils with high electrical conductivity can lead to the effect that the measured in-phase component is also a response of conductivity and not only magnetic susceptibility [4]. This effect increases with increasing frequency and coil distance. Nevertheless, ref. [4] concluded that EMI surveys remain a viable option for subsurface prospection in highly saline conditions.

In this work, we aimed to test the applicability of frequency-domain-EMI to the prospection of wooden archaeological targets in the difficult environment of tidal flats. For this purpose, we focussed on a known 17th century wooden wreckage west of the tidal island of Hooge (North Frisia, Germany), which was found in 2017 on the sandbar "Japsand" (Figure 1). This sandbar, with the characteristics of a barrier island, is subject to strong sedimentary dynamics of the outer coastline and shifts steadily eastwards. For the period 1947–2001, map comparisons show a retreat of the western coastline by c. 20 m per year [20].

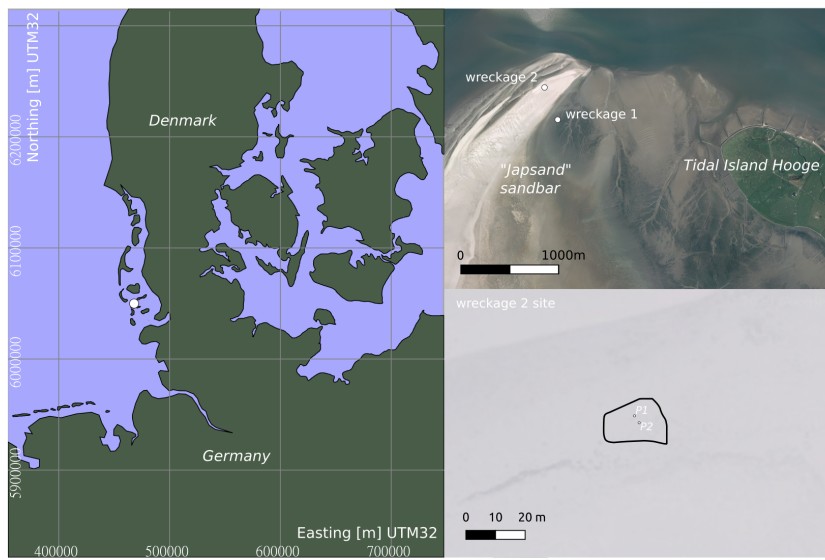

**Figure 1.** Position of the investigation area (white dot). The black line down the right outlines the measurement area, P1 and P2 are fixed datum points defining the position of the archaeological target. (Aerial foto right: © LVermGeo SH/ALSH, 2013).

The questions we are dealing with in this case study are as follows

- Does EMI resolve thin wooden wreck parts?
- How is the contrast in seawater saturated/partly saturated by the sandy sediment?
- What are the difficulties for EMI when dealing with seawater environments and how can we address them?

The test site was chosen at a silted, articulated slab of planking measuring about 8 × 3 m. It emerged on top of the higher parts of Japsand in May 2017 as a consequence of coastal erosion. It was entirely built of oak and up to 7 strakes held together by partially joggled frames were preserved. The wreck-piece features the curvature of the hull at the turn of the bilge. The thickness of the planks varies between 3 and 4.5 cm. The wood was extremely well preserved, as it showed no signs of biological deterioration caused by marine borers or other organisms. It was also not in an advanced waterlogged state (i.e., internal wood-cell dissolution through permeation when exposed in a water-column over a longer period of time). This shows that the wreckage must have been covered by sediments for most of the time since the ship's loss. Shortly after the wreckage was excavated, sampled, and documented by the State Archaeology Department of Schleswig-Holstein (ALSH) in May 2017, it silted in once again, as was reported by ALSH's local contact Dirk Bienen-Scholt, who regularly visits this area. Both the wreck itself and the voluntary site-monitoring by the latter made the wreck an ideal target for this case study.

## 2. Materials and Methods

### 2.1. Electromagnetic Induction Survey

Frequency-domain electromagnetic induction (EMI) devices use electromagnetic waves generated in a transmitter and recorded in one or several receiver coils to create

maps of electromagnetic subsoil properties at different sensing depths. The transmitter coil emits a 'primary' harmonic oscillating electromagnetic wavefield (kHz frequency range). This field induces eddy currents in the subsoil that depend on the electrical conductivity of the soil. These oscillating eddy currents generate a 'secondary' field, which in super-position with the primary field is recorded at the receiver coils. Based on this, the EMI method measures the apparent electrical conductivity of the soil and the so-called in-phase property, which is a function of the magnetic susceptibility (see e.g., [21]) for the LIN case. For higher induction numbers, the in-phase data is also dependent on the conductivity, as mentioned in the introduction. The penetration depth of the EMI method depends on the device configuration, namely the signal frequency, the transmitter-receiver distance and orientation, the measurement height above ground, but also on the soil properties them-selves for non-LIN cases. In this study, we used a CMD Mini-Explorer by GF Instruments (Figure 2a). The device consists of one transmitter and three receiver coils. The planes of the coils were oriented horizontally (horizontal coplanar—HCP). The distance between the transmitter and receivers were 0.32, 0.71, and 1.18 m leading to effective theoretical depths of exploration of 0.5, 1.0, and 1.8 m in HCP mode. Further details on the method and the device can be found in [14]. The measurements were performed with a 10 Hz sampling frequency. The area was covered by randomly walked W–E-orientated and N–S-orientated tracks with an average spacing of about 0.5 m between the tracks. The N–S tracks were used to derive the instrument drift. An RTK-GPS (Stonex S9i) was connected to the device using a fixed fiberglass rod at a distance of 2 m from the coils, minimizing the influence of the electronics on the data. Furthermore, the handheld control device was connected via Bluetooth and carried by a second person at a few meters distance for the same reason. Measurement time to cover the area around the wreckage was about 15 min, covering 260 m². Processing of the data included the following steps:

- Correcting RTK-GPS positions for each individual coil pair center offset.
- Bandpass inline spatial filtering with the corner frequencies 0.0001 (1/samples) and 0.2 (1/samples).
- Drift (after [22]) and height correction of the data. The latter was applied, because small changes in instrument height and distance to the groundwater level create offsets in the data that are in the magnitude of the target signal itself. These changes are caused by the sand ripples' topography and the height error due to carrying the instrument by hand. The height correction is performed by determining a correlation trend between GPS antenna height and either apparent conductivity or IP value. This linear trend is then corrected to the mean of the data values. The data of each coil distance can thus be assumed to correspond to a distinct depth level after this correction.
- Gridding and smoothing of the data using a grid increment of 0.25 m and Gaussian image filter with 0.5 m half width.

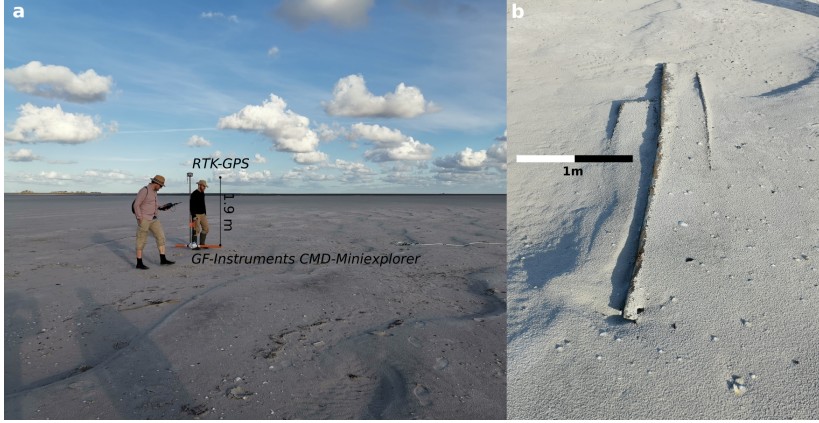

**Figure 2.** (**a**) The used EMI device and field setup at the site. (**b**) Parts of the wreck remains that were visible on the surface in April 2020.

Due to the short available low tide time window and the long walking distance from the island to the site, an ERT calibration profile providing specific conductivities as proposed by [18] could not be performed. We thus rely on qualitative interpretation of the data. The results are, therefore, described as apparent conductivities and IP-values, including the uncertainty of calibration and the uncertainty of the LIN approximation. A proper inversion of data for true conductivity would require calibrated data and is thus not performed.

### 2.2. Excavation

The targeted slab of planking (i.e., Figure 3, wreck part 2) is actually part of a larger wreckage scattered across Japsand and the tidal mud-flats to the east, where a new tidal creek has formed recently, which can be probably linked to the wreck's exposure. As inferred by the break-line along the plank-scarphs, as well as the result of the dendrochronological analysis, providing a precise terminus post quem of 1609 for wreck part 1 and 1608 for wreck part 2, it can be confidently inferred that the wreck-pieces were of the same wreck and once connected, despite being discovered ca. 400 m apart [23]. It is not clear whether the breakage occurred already at the time of the ship accident or thereafter. In fact, the shipwreck may have occurred at a different location, and the wreckage simply drifted onto the same sand by the same current. Unlike wreck part 1, wreck part 2 has not been encountered in a fully exposed state, which would have been a precondition for a piece to break off in more recent times. Owing to its find location higher up on the Japsand, which is only occasionally entirely inundated during a spring tide or a westerly gale, wreck part 2 has become covered by sediment again. This directly led to its prolonged in situ preservation: The sediment cover provides protection, as the wreck is no longer exposed to mechanical stresses caused by tidal currents and is also shielded from the impact of marine borers, such as the teredo navalis. The first wreck piece, however, was swiftly reclaimed by the sea shortly after its discovery in February 2017.

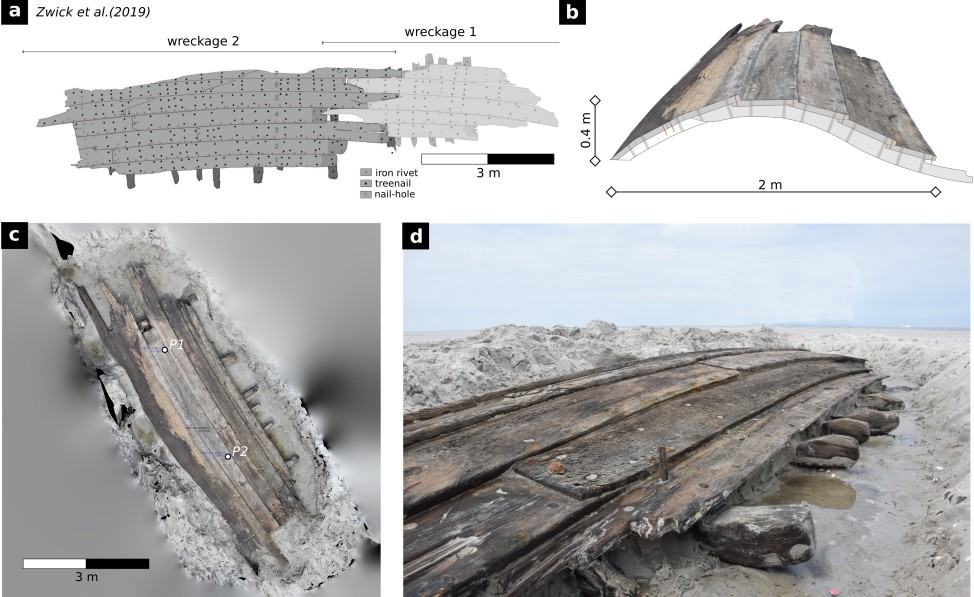

**Figure 3.** (**a**) Drawing of the wreck after the excavation results [23]. The excavation results act as ground truthing data for the application test. The illustrated part of the wreckage is almost completely buried in the refilled excavation pit today. (**b**) Cross-section of wreck part 2 generated from an SfM-model and manual annotations (**c**) Structure-from-Motion (SfM) orthophoto of wreck part 2 generated with Agisoft Megashape. (**d**) The excavation of wreck part 2 in May 2017 (ALSH).

## 3. Results

Figure 4 shows the effect of the applied data processing steps. Especially the correlation between system height and apparent conductivity (as illustrated for the coil distance of 0.72 m and quadrature component in Figure 4f) is noticeable. The step successfully removes the topography/surface sand ripple structures from the data (Figure 4c,d in comparison to Figure 4e).

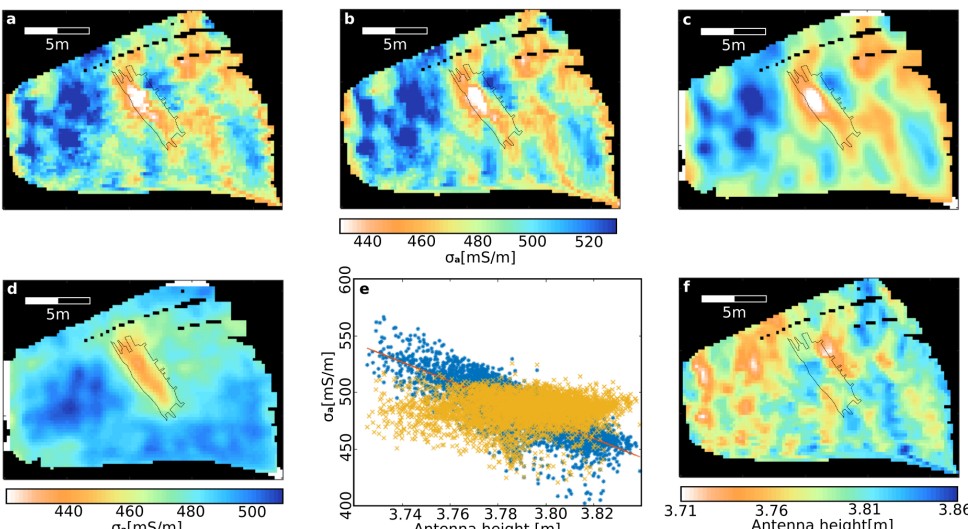

**Figure 4.** Processing steps illustrated for the apparent conductivity map of coil distance 0.72 m. (**a**) Gridded raw data, (**b**) data corrected for coil pair center position, (**c**) inline spatial filtering, (**d**) drift and height corrected data. (**e**) Shows the applied height correction and the correlation of antenna height and apparent conductivity. (**f**) Linearly interpolated GPS antenna height. The position of the wreckage, as found in the prior excavation, is outlined by a dotted line.

Figure 5a–c shows the mapped apparent conductivity, calculated with the LIN approximation from non-calibrated quadrature data for three coil distances. The images also show the position of the wreckage as a black dashed line. The wreckage shape is clearly visible at least at the two largest coil distances. In Figure 5b, a solid black line indicates the position of the test profile shown in Figure 5d. The profile illustrates the EMI signal form together with the position and cross-section of the wreckage. The curvature of the ship hull is displayed as a dashed line and correlates with the apparent conductivity signal shape.

Figure 6a–c shows the mapped in-phase (IP) component for three coil distances. The images also show the position of the wreckage as a black dashed line. The outline of the wreckage is only visible in parts and less clear as in the conductivity maps. Figure 6d illustrates the effect of conductivity on the in-phase component, showing a linear trend with a $2\sigma$ (95%) confidence interval.

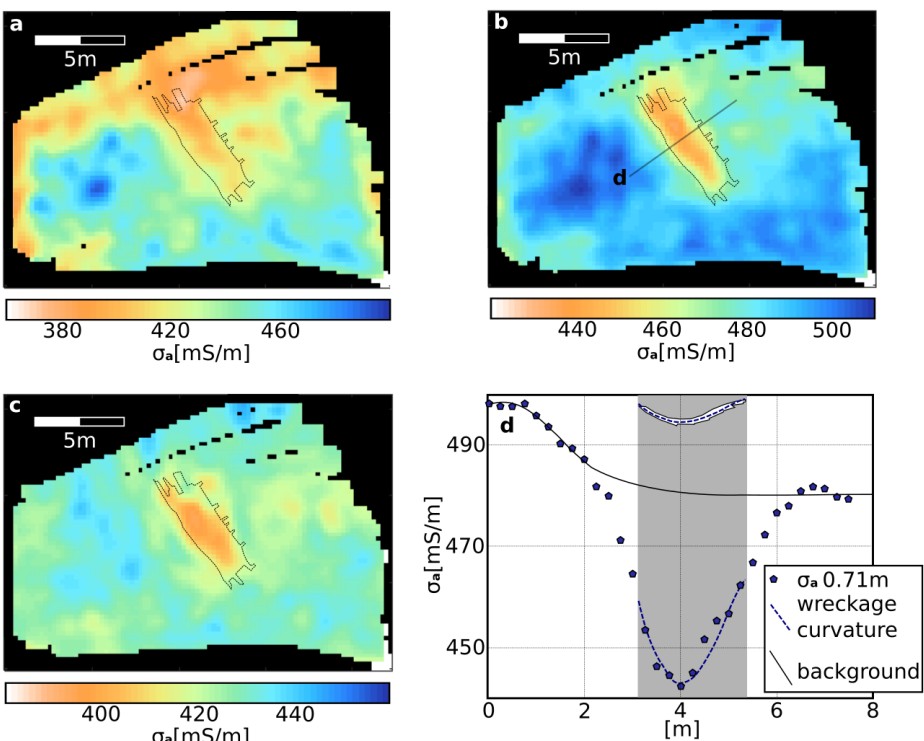

**Figure 5.** Resulting apparent conductivity maps for coil distances 0.32 m (**a**), 0.71 m (**b**), and 1.18 m (**c**). The position of the wreckage is outlined by a dotted line. (**d**) Shows an example profile of coil distance 0.71 m (shown in **b**) together with the position and cross-section of the wreckage. The curvature of the ship hull is displayed as a dashed line.

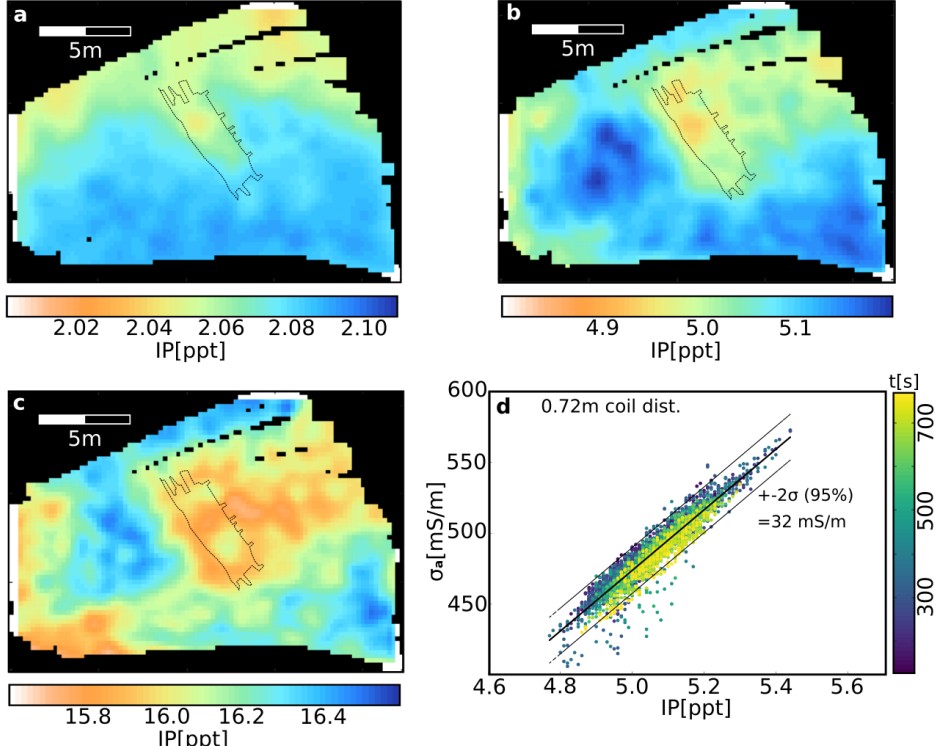

**Figure 6.** Resulting IP maps for coil distances 0.32 (**a**), 0.71 (**b**), and 1.18 m (**c**). The position of the wreckage is outlined by a dotted line. (**d**) IP vs. quadrature component raw data for coil distance of 0.72 m.

## 4. Discussion

The results in Figure 5 show that FD-EMI surveys provide the potential to image wooden archaeological objects, such as the presented ship wreckage, in shallow depths in tidal flat surroundings. Nevertheless, several effects, limitations, and benefits need to be discussed.

The results show that the contrast and data dynamics in terms of conductivity are quite low. The amplitude of the wreckage signal is about 10% of the background signal (regarding the coil distance of 0.72 m, Figure 5d), which is in the same order of magnitude as the drift and height effects. Though data processing and final interpretation needs to be performed with caution. The dynamic range of the data is about 100 mS/m (comparable to the results of [4]). Ref. [12] investigated the contrasts of geoelectric sounding and ground-penetrating radar for spruce and oak wood of different water saturation in a laboratory experiment. They showed that the electrical resistivity showed the highest contrast in comparison to seismic velocity and dielectric permittivity in freshwater, depending on the surrounding material and structural orientation of the wood. However, in a saline environment, such as the present, the contrast is probably much lower due to the decreased resistivity of the subsoil surrounding and the wood, which is indicated by the presented results but needs to be verified by inversion of true resistivity. However, to be able to discuss contrast quantitatively and apply proper inversion schemes, we would need to account for the erroneous assumption of LIN-approximation applied in saline environments and the lack of a suitable device calibration curve, both contributing to the limited dynamic. Furthermore, conductivity measurements severely depend on the soil temperature (especially of the pore fluid) and, of course, the water saturation of the soil. In tidal flat areas, these two parameters change in a very short time window with the changing tide. Thus, device drift, soil temperature, and drift due to changing water saturation superimpose to a generalized data drift curve and cannot be separated. The effect of the changing water level can be estimated using a dataset that was recorded on the nearby island of Sylt (Figure 7). Apparent resistivities were measured in the tidal flats at the northern coast of the island from 3 to 4 August 2021, covering nearly two tidal cycles. The data was acquired using a fixed inverse Schlumberger array with electrodes mounted on the seafloor (distance 0.5 m for potential electrodes and 2.5 m for current electrodes). The apparent resistivities were measured every 60 s using a RESECS unit. Usually, the electrodes are supposed to be point-shaped and to be located at the surface of a homogeneous half-space. However, as soon as the electrodes get overflooded, this assumption is not satisfied anymore. Electrodes are then part of a limited full space causing a change in the field geometry. This effect was corrected by utilizing an analogy to optical mirror images following [24]. The data shows that during high tide we see a fully saturated subsoil with clear resistivity minimum. Changes are continuous due to the changing water level above ground. The maximum apparent conductivity of the first tidal cycle is about 1800 mS/m. During low tide, we observe that the conductivity values reach a plateau starting at about 830 to 780 mS/m. We expect that the water level has sunken below the depth of investigation and though we only observe a seeping and slow drying of the topsoil, resulting in a change of conductivity of about 50 mS/m in 7.5 h. In a measurement window of one hour during low tide, this means a change of conductivity of about 7 mS/m, an effect that is superimposed on the device drift and influences long-lasting ERT measurements or coring/EC-logging for calibration attempts. A fast four electrode vertical electric sounding would be appropriate but of course integrates over a large subsoil volume. Furthermore, the water conductivity changes by about 3% for each 1 °C change in water temperature. In the regarded region, the water temperature changes by 3° in a daily manner [25]), which translates into a change of conductivity (at low tide at different times a day) of 60 mS/m.

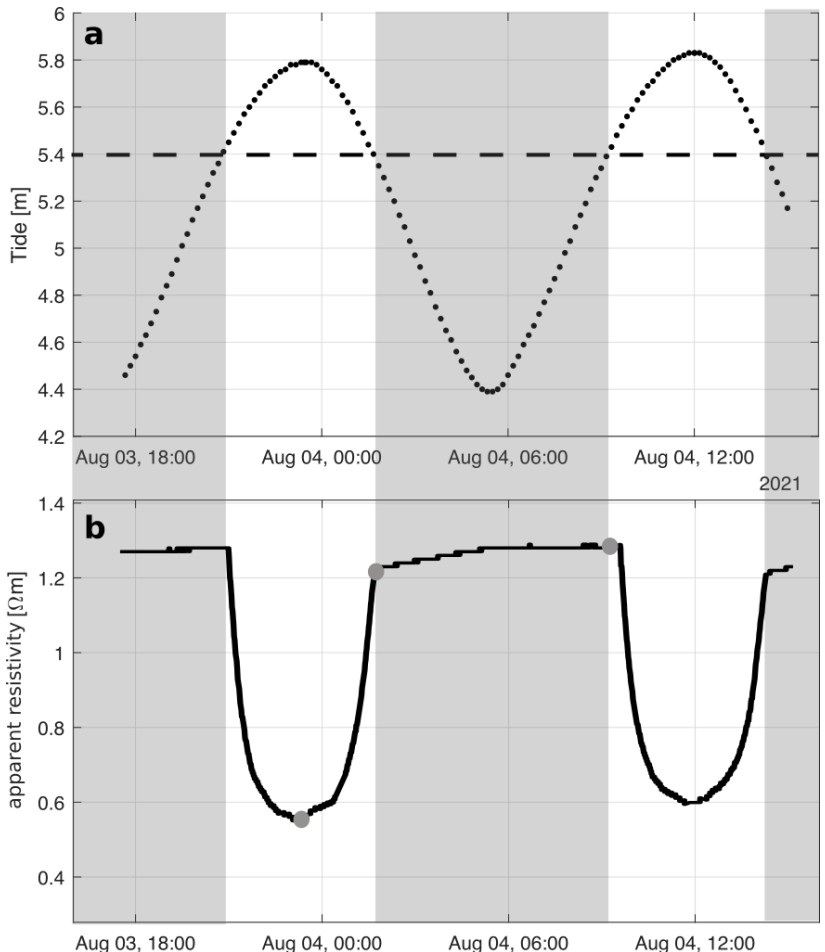

**Figure 7.** Tide gauge level (**a**) and near-surface apparent resistivity (**b**) measured at the coast of the island of Sylt (Königshafen) starting on 3 August until 4 August 2021, covering nearly two tidal cycles. The data in b was acquired using a fixed inverse Schlumberger array in the tidal flat area at the northern coast of the island of Sylt.

Figure 6d reveals a correlation trend between IP and conductivity data as already described and expected by [4], whereas they observed a deviation from a linear relationship at the start and end of their survey. As both datasets (Figures 5 and 6) show differences, especially in the long-wavelength features, when compared visually, we expect the differences to be covered by the variation from the linear trend (the fitted linear regression in Figure 6d shows a 95% confidence interval of 32 mS/m). The image quality of the wreckage in terms of IP is lower, indicating a lower contrast. However, EMI still benefits from fast measurement progress, easy handling in terms of weight, and depth resolving abilities, if calibrated. The absence of calibration data prevents performing a proper inversion on the data to access conductivity-depth models. If calibration data is available, non-LIN forward modeling must be used for inversion and can lead to depth information of the targets and stratigraphic layers. In the presented case, calibration data is not available due to the short available low tide time window. The assessment of imaging properties and depth sensitivity is possible anyway. In terms of imaging capabilities, we see a distinct signal of the investigated wreckage, clearly outlining its shape in two depth ranges (0.72 and 1.18 m coil distances) (Figure 5b,c). Because the investigated target has a curved shape, continuously covering a depth range of about 0.5 m, which correlates well with the EMI signal shape (Figure 5d), we can assume a sufficient depth sensitivity. Nevertheless, in terms of general sensing depth, the used device probably covers only the first 1 m of the subsoil, as [4] showed that the depth of exploration is significantly reduced in high conductive areas (tidal flats). In the tidal flats of North Frisia, however, most of the

medieval archaeology is contained in the first meter [3], making EMI a feasible method for archaeological prospection at remote places, such as the beaches and barrier islands of the Wadden Sea, which can be regarded as natural wreck archives (cf. Figure 8; Ref. [26]). Incidentally, these are also the locations where most of the historically recorded shipwrecks occurred [27].

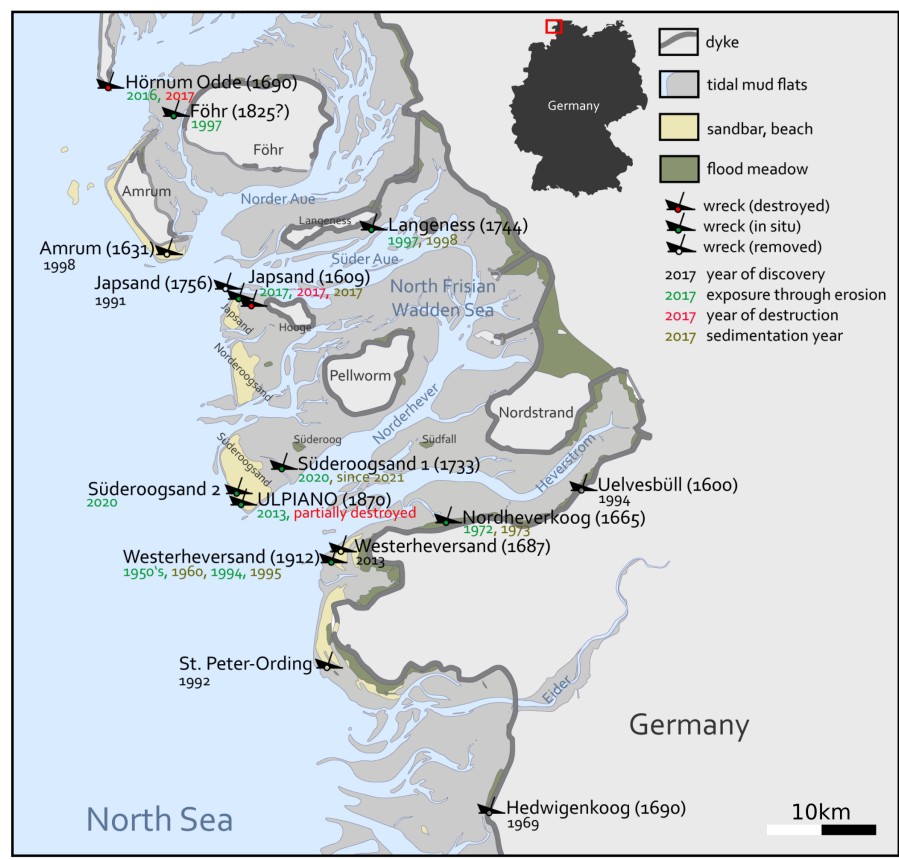

**Figure 8.** This overview map shows all known historical shipwrecks (dates in brackets) in Schleswig-Holstein's Wadden Sea, which were subject to an archaeological investigation. The overview demonstrates that most wrecks were affected by erosion and sedimentation. Wrecks exposed by erosion have either been destroyed or removed, or covered by sediments again after a short period of time (graph: Daniel Zwick/base map: LKN.SH).

## 5. Conclusions

We presented a feasibility study by testing the potential of an EMI survey to identify a wooden archaeological target in the tidal flat area of the North Sea. The target is a part of a 17th century ship, which was archaeologically investigated prior to this study. To conclude on our initially posed questions, the study showed that

- EMI is able to resolve wooden archaeological objects in tidal flats.
- The investigated wooden object shows an adequate contrast in terms of conductivity, if several effects of similar magnitude, such as device height, are integrated into the data processing.
- A sufficient depth sensitivity is shown in the near surface, but well adapted calibration efforts need to be developed to obtain apparent conductivities suitable for inversion, to deal with the highly conductive environment, and to include the tide-influence on conductivity in the tidal flat subsoil.

The study highlights the great potential of EMI for both in situ heritage management and archaeological research in the Wadden Sea. As far as this case study can show, high-potential areas for further pieces of wreckage could be identified through EMI-detected

anomalies. Moreover, EMI surveys in conjunction with predictive modeling of coastal erosion processes have the potential to improve the planning of rescue excavations well before archaeological sites are immediately threatened by coastal erosion. Once exposed by erosion, the site is usually subject to the destructive influences of tidal currents and the surf zone. Thus geophysical methods that are capable of contributing to an "early-warning system" for potential new wreck discoveries would be of great benefit in view of the complexity of wreck-research and the preparation required.

**Author Contributions:** Conceptualization, D.W.; methodology, D.W. and D.Z.; software, D.W.; validation, D.Z.; investigation, D.W., B.S.M., R.B., D.B.-S, S.F. and E.E.; writing, D.W., D.Z. and B.S.M.; review and editing, R.B.; visualization, D.W. and D.Z.; project administration, D.W., B.S.M., R.B. and D.B.-S.; funding acquisition, D.W. and B.S.M. All authors have read and agreed to the published version of the manuscript.

**Funding:** This study was funded by the DFG project HA 7647/1-1 and the ROOTS Cluster of Excellence funded by the German Research Foundation (EXC 2150–390870439).

**Informed Consent Statement:** Not applicable.

**Data Availability Statement:** The data is available under CC-BY 4.0 license at www.pangaea.de, PDI-30249.

**Acknowledgments:** The authors would like to thank Detlef Schulte-Kortnack and Clemens Mohr for their extensive technical help. We would also like to thank the AXIO-NET PED-Service for providing their RTK-correction data and Stefanie Klooß (ALSH, State Archaeology Department of Schleswig-Holstein) for her ongoing support of our work in the North Frisian Wadden Sea.

**Conflicts of Interest:** The authors declare no conflict of interest.

## Abbreviations

The following abbreviations are used in this manuscript:

| | |
|---|---|
| FD-EMI | Frequency-domain electromagnetic induction |
| EMI | Electromagnetic induction |
| LIN | Low Induction Number |
| ERT | Electrical Resistivity Tomography |
| RTK-GPS | Real-Time Kinematic Global Positioning System |
| DFG | German Research Foundation |

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
