# Peer review of "Multi-Coil FD-EMI in Tidal Flat Areas: Prospection and Ground Truthing at a 17th Century Wooden Ship Wreckage"

_remotesensing, doi:10.3390/rs14030489_

Round 1
Reviewer 1 Report
Remote geophysical sensing methods are most welcome to archaeological and cultural heritage research.
GPR, magnetic, electrical, EMI are used to such purpose employing various advanced algorithms in a 2D or 3D mapping.
The presented study of using multi-coil frequency domain electromagnetic (FD-EMI) prospection of a wooden ship wreckage from the 17th century is a worth investigation.
Delefortrie et al. (2014) have concluded in the past that EMI surveys remain a viable option for subsurface prospection.
However variables concerning conductivity agents surely affect the expected result.
1)The authors should become more clear quantitatively on the resolution regarding depth, dimensions, salinity, semi-covered wreckage by sediment, EM conductivity of hydrated wood etc
2)Any experimental testing if made should be reported.
3)As EM prospection is an well-known prospecting technique some relevant early works must be mentioned in EM, GPR methods applied to case studies:
Adriaan De Man, João Tiago Tavares and Jorge Carvalho (2017) gpr and electromagnetic induction surveys at the hilltop site of ul (oliveira de azeméis, portugal) Mediterranean Archaeology and Archaeometry, Vol. 17, No 1, , pp. 179-188 DOI: 10.5281/zenodo.292622
S.Aydıngün, Y.C. Kurban, C.Ç. Yalçıner, A. Büyüksaraç, E. GündoÄŸdu, E. Altunel (2020) high-resolution ground penetrating radar investigation of yerebatan (basilica) cistern in istanbul (constantinople) for restoration purposes. Mediterranean Archaeology and Archaeometry Vol. 20, No 3, pp. 13-26 DOI: 10.5281/zenodo.3930402
Büyüksaraç, E. Eser, Ö. BektaÅŸ, B. Akay, S. KoÅŸaroÄŸlu (2012) SURFACE GEOPHYSICAL INVESTIGATIONS AND PRELIMINARY EXCAVATIONS AT THE DIVRIGI CITADEL, SIVAS (TURKEY), Mediterranean Archaeology and Archaeometry, Vol. 12.1, No. 1, pp. 117-128.
Author Response
On behalf of the authors, I would like to thank the reviewers for their effort and constructive comments on our manuscript.
Please find here our response to the comments of
REVIEWER 1:
Remote geophysical sensing methods are most welcome to archaeological and cultural heritage research.
GPR, magnetic, electrical, EMI are used to such purpose employing various advanced algorithms in a 2D or 3D mapping.
The presented study of using multi-coil frequency domain electromagnetic (FD-EMI) prospection of a wooden ship wreckage from the
17th century is a worth investigation.
Delefortrie et al. (2014) have concluded in the past that EMI surveys remain a viable option for subsurface prospection.
However variables concerning conductivity agents surely affect the expected result.
1)The authors should become more clear quantitatively on the resolution regarding depth, dimensions, salinity,
semi-covered wreckage by sediment, EM conductivity of hydrated wood etc.
Answer: The dataset does not allow proper depth resolution quantitative analysis, because it is not calibrated. As explained in the manuscript (from line 76
in the introduction; from line 164; and from line 227). The calibration of the CMD Explorer was done at 50mS/m (http://www.gfinstruments.cz/index.php?menu=gi&smenu=iem&cont=cmd_&ear=ts)
and is only valid for valid LIN-Approximation. The conductivities in the saltwater saturated tidal flats are far from this range. We explained in the
discussion, line 253 how calibration in the field can be done in the future, involving VES Measurements or direct push EC-Logging.
However, a full solution modelling study could give estimate on the depth range and possible resolution but would not provide quantitative results for the
dataset. Did you mean such kind of a modelling study? Conductivtiy values for hydrated, and especially dergraded archaeological wood in salt water are not
available as far as I know. A study involving archaeological wood in freshwater is discussed from line 64 or 219.
2)Any experimental testing if made should be reported.
Answer: What do you mean? All experimental work is described in the manuscript, involving EMI-data acquisition and VES data acquisition.
3)As EM prospection is a well-known prospecting technique some relevant early works must be mentioned in EM, GPR methods applied to case studies:
Answer: In the introduction we have mentioned several works dealing with prospection in tidal flat areas, e.g. Delefortrie (2014), Schwardt (2020),
Fediuk (2020) in terms of ERT; Missiaen (2010/17) in terms of seismics. Also general EMI works are mentioned in line 71. As GPR is not feasible in
saltwater areas like tidal flats as mentioned in the introduction, we do not see any benefit in mentioning GPR works dealing with completely different
environments.
Reviewer 2 Report
This manuscript describes a geophysical experiment evaluating the practicality of a 3-configuration loop-loop EMI sensor for characterizing a wooden buried archaeological target in the intertidal zone. The text is well structured and well written. All the applied methodology is sound and adapted to the state of the collected data set. I am therefore in favor for a publication in Remote Sensing. Please find below some minor suggestions/comments.
Line 89-90: “it would be advisable to invert raw data of the EMI etc..”
This statement is rather unexpected from the theoretical point of view. Converting Hs/Hp data into apparent conductivity corresponds to a change of data space. Accordingly, I don’t see how this can be an obstacle for inversion even if the chosen data space presents some complex features. Indeed, within an inversion procedure, the forward modelling does take this transformation into account (for example in gradient-based inversion approach, the jacobian has to be derivated differently if we consider Hs/Hp data or apparent (LIN or not) conductivity data). Changing the data space yields a change of the preconditioning of the system to be solved in the framework of an optimization/inversion problem. And for this aspect, considering LIN or robust apparent conductivity actually gives more benefits as the related data space is by definition not (or poorly for LIN conductivity) affected by acquisition parameters like frequency, coil separation, geometry, height, etc... In other words, the associated question is "Do we want a data point inherently unweighted because it has been recorded with, for instance, a certain coil spacing?" The response is for me, “no”. The most popular and classical example of such data space change and inversion-preconditioning is the one routinely performed for the ERT method (voltage -> apparent resistivity ->inversion). In conclusion, despite the limitation of LIN conductivity for direct interpretation, their inversion does not suffer of critical bias in comparison to other kinds of “data format”.
Line 132 “In-phase is a function of magnetic susceptibility (only)”: Here, it should be recalled that this is true only for low induction numbers, otherwise in-phase data is also dependent on the conductivity (as rather well known from classical textbooks, e.g., Ward and Hohmann ,1988).
I think a legend should be added to Figure 5d, to describe the different curves that are shown.
Line 224-226: “low contrast of conductivity”. The loop-loop EMI method has a poor sensitivity to resistivity anomalies. This means that a large resistivity contrast in the subsurface may be however characterized by a small anomaly of the data. The obvious way to correct this effect is to perform an inversion. Because no inversion is provided in this study, the authors cannot really assert that the subsurface resistivity contrast caused by the buried wood is small.
Line 279-282 “by about 25 %”: The variability of depth sensitivity with subsurface conductivity should be quantified specifically for each instrument, as this feature strongly depends (and increases) with the frequency and the coil-spacing. Especially because this study considers smaller coil-spacings than in Delefortrie et al, (2014), this value of 25% should be presented more cautiously.
Best regards
Author Response
On behalf of the authors, I would like to thank the reviewers for their effort and constructive comments on our manuscript.
Please find here our response to the comments of
REVIEWER 2
This manuscript describes a geophysical experiment evaluating the practicality of a 3-configuration loop-loop EMI sensor for characterizing a wooden
buried archaeological target in the intertidal zone. The text is well structured and well written.
All the applied methodology is sound and adapted to the state of the collected data set.
I am therefore in favor for a publication in Remote Sensing. Please find below some minor suggestions/comments.
1) Line 89-90: “it would be advisable to invert raw data of the EMI etc..”
This statement is rather unexpected from the theoretical point of view. Converting Hs/Hp data into apparent conductivity corresponds to a change of data space. Accordingly, I don’t see how this can be an obstacle for inversion even if the chosen data space presents some complex features. Indeed, within an inversion procedure, the forward modelling does take this transformation into account (for example in gradient-based inversion approach, the jacobian has to be derivated differently if we consider Hs/Hp data or apparent (LIN or not) conductivity data). Changing the data space yields a change of the preconditioning of the system to be solved in the framework of an optimization/inversion problem. And for this aspect, considering LIN or robust apparent
conductivity actually gives more benefits as the related data space is by definition not (or poorly for LIN conductivity) affected by acquisition parameters like frequency, coil separation, geometry, height, etc... In other words, the associated question is "Do we want a data point inherently unweighted because it has been recorded with, for instance, a certain coil spacing?" The response is for me, “no”. The most popular and classical example of such data space change and inversion-preconditioning is the one routinely performed for the ERT method (voltage -> apparent resistivity ->inversion). In conclusion, despite the limitation of LIN conductivity for direct interpretation, their inversion does not suffer of critical bias in comparison to
other kinds of “data format”.
Answer: I totally agree with your arguments, if we are talking about a scaling of data space, if linear or not. The statement you mention is cited from
Benech et al. (2016); from my understanding, it targets the conductivity dependance of the IP-component which results in a trade-off between IP and
conductivity if used seperately. Nevertheless, the sentence is probably a bit misleading in this paragraph.
2) Line 132 “In-phase is a function of magnetic susceptibility (only)”: Here, it should be recalled that this is true only for low induction numbers,
otherwise in-phase data is also dependent on the conductivity (as rather well known from classical textbooks, e.g., Ward and Hohmann ,1988).
Answer: I changed that paragraph, recalling the dependance of IP on conductivity as well.
3) I think a legend should be added to Figure 5d, to describe the different curves that are shown.
Answer: I added a legend. The curves were truely a bit confusing.
4) Line 224-226: “low contrast of conductivity”. The loop-loop EMI method has a poor sensitivity to resistivity anomalies. This means that a large
resistivity contrast in the subsurface may be however characterized by a small anomaly of the data. The obvious way to correct this effect is to perform
an inversion. Because no inversion is provided in this study, the authors cannot really assert that the subsurface resistivity contrast caused by the
buried wood is small.
Answer: This is of course only an indication. I added the sentence "but needs to be verified by inversion of true resistivity".
5) Line 279-282 “by about 25 %”: The variability of depth sensitivity with subsurface conductivity should be quantified specifically for each instrument,
as this feature strongly depends (and increases) with the frequency and the coil-spacing. Especially because this study considers smaller coil-spacings
than in Delefortrie et al, (2014), this value of 25% should be presented more cautiously.
Answer: Sure, this was only an estimate but obviously not supported by data. I removed the value from the text.
Reviewer 3 Report
This is not my area of specialty but I don't see any problems with the interpretation nor data. I do strongly recommend breaking up the narrative into paragraphs, though.
Author Response
On behalf of the authors, I would like to thank the reviewers for their effort and constructive comments on our manuscript.
Please find here our response to the comments of
REVIEWER 3
This is not my area of specialty but I don't see any problems with the interpretation nor data.
I do strongly recommend breaking up the narrative into paragraphs, though.
Answer: I tried to add paragraphs where possible